# Expression of CD44v6-Containing Isoforms Influences Cisplatin Response in Gastric Cancer Cells

**DOI:** 10.3390/cancers12040858

**Published:** 2020-04-02

**Authors:** Carla Pereira, Daniel Ferreira, Nuno Mendes, Pedro L. Granja, Gabriela M. Almeida, Carla Oliveira

**Affiliations:** 1i3S - Instituto de Investigação e Inovação em Saúde, Universidade do Porto, 4200-135 Porto, Portugal; carlap@ipatimup.pt (C.P.); daniel.ferreira@i3s.up.pt (D.F.); nmendes@ipatimup.pt (N.M.); pgranja@ineb.up.pt (P.L.G.); 2IPATIMUP - Institute of Molecular Pathology and Immunology, University of Porto, 4200-135 Porto, Portugal; 3INEB - Instituto de Engenharia Biomédica, Universidade do Porto, 4200-180 Porto, Portugal; 4ICBAS - Institute of Biomedical Sciences Abel Salazar, University of Porto, 4050-313 Porto, Portugal; 5FMUP - Faculty of Medicine of the University of Porto, 4200-319 Porto, Portugal

**Keywords:** stomach cancer, chemotherapy, CD44 variants, apoptosis, cell population dynamics

## Abstract

CD44v6-containing isoforms are frequently de novo expressed in gastric cancer (GC). Whether CD44v6 has a central role in GC transformation and/or progression, whether it conditions response to therapy or whether it is only a bystander marker is still not known. Therefore, we aimed to clarify the role of CD44v6 in GC. We generated GC isogenic cell lines stably expressing CD44s or CD44v6 and tested them for different cancer hallmarks and response to cisplatin, and we further confirmed our findings in cells that endogenously express CD44v6. No correlation between overexpression of CD44v6 and the tested cancer hallmarks was observed, suggesting CD44v6 is not a driver of GC progression. Upon cisplatin treatment, CD44v6+ cells survive better and have lower apoptosis levels than CD44v6− cells, possibly due to concomitant activation of STAT3 and P38. In co-culture experiments, we discovered that CD44v6+ cells are involved in GC cell overgrowth after cisplatin treatment. In conclusion, we show that CD44v6 expression increases cell survival in response to cisplatin treatment in GC cells and that these cells override CD44v6-negative cells after cisplatin-treatment. This suggests that tumor expression of CD44v6-containing variants may condition the outcome of GC patients treated with chemotherapy.

## 1. Introduction

CD44 is a ubiquitous membrane receptor that was first described for its role as an organ-specific, lymphocyte homing, cell surface molecule [1]. It has since been associated with a large number of key cellular adhesion processes, such as homotypic and heterotypic cellular adhesion [2,3]. Additionally, CD44 mediates cell to extracellular matrix (ECM) adhesion, as shown by its function as the major receptor for hyaluronic acid, a major component of the ECM [4], and also by its ability to interact with fibronectin [5] and collagen [6]. CD44 has also been described as a co-receptor with a pivotal role in intracellular signaling, mediating key aspects of cellular behavior such as motility and cell survival [7,8,9].

The human CD44 gene (NG_008937) encodes a polymorphic group of transmembrane glycoproteins generated by alternative splicing. The standard CD44 isoform (CD44s) includes only the constitutive exons, while the variant isoforms (CD44v) contain one or more variable exons (in addition to the constitutive ones) [7,9,10,11].

Regarding gastric tissue, we have previously shown that CD44s is widely expressed in both normal and diseased gastric epithelial cells [12]. In contrast, CD44v6-containing isoforms are de novo expressed in stomach premalignant and malignant lesions and in ~70% of all gastric cancers [12], the third leading cause of cancer related mortality worldwide [13]. Aberrant expression of CD44v isoforms has been associated with several cancer-related features like invasion and metastization [10,14,15]. Moreover, CD44 and its CD44v isoforms are expressed as surface markers of cancer stem cells (CSCs), influencing key CSC-associated properties such as tumor initiation, self-renewal, metastasis and chemoresistance [10,16,17,18]. Indeed, increased expression of CD44 has been observed in trastuzumab resistant gastric [19] and breast [20] cancer cells, as well as in paclitaxel resistant ovarian cancer cells [21]. Likewise, expression of some CD44v has been associated with chemoresistance, like CD44v6 in pancreatic/prostate/colon cancer cells [22,23,24] and CD44v9 in gastric cancer cells [25]. Moreover, an association between increased CD44 or CD44v6 tumor expression and worse overall survival OS in gastric cancer (GC) patients has previously been reported [26,27,28]. In sum, much is known about the molecule CD44v6 (in general) and several studies suggest that it can have a function in GC. However, it is still not known whether the observed de novo expression of CD44v6 has a central role in the transformative process and/or progression of GC, if it conditions response to therapy or if it is only a bystander marker resulting from all or some of those events. Therefore, in the present work, we aimed to clarify the role of CD44v6 using both an isogenic model of GC cells stably expressing CD44s or CD44v6, and GC cells that endogenously express CD44v6.

## 2. Results

### 2.1. CD44v6 Overexpression is not A Driver of GC Progression

In an attempt to clarify the role of CD44v6 overexpression in GC, we generated isogenic cell lines by stably expressing in a CD44-null MKN74 GC cell line (Appendix A), the following sequences: (i) a CD44v6-containing isoform highly expressed in GC cells (GP202 cell line, Appendix A) [12], from now on mentioned as MKN74_CD44v6 cells (Figure 1A); (ii) the CD44s isoform (MKN74_CD44s cells, Figure 1A), or; (iii) the empty plasmid (MKN74_Mock cells, Figure 1A). Characterization of the generated cells confirmed the expression of the intended transcripts at the transcript and protein levels, as well as its localization at the cell membrane (Figure 1B–D). Moreover, the CD44v6 expression levels, obtained here, are physiologically relevant since they are similar to those endogenously expressed in the GP202 cell line [Pereira, C.; Ferreira, D.; Almeida, G.M.; Oliveira, C. Personal observation, 2018.]. These three isogenic cell lines were further characterized and no differences in growth rate, invasion, cell-cell aggregation or migration capacity were observed between them (Figure 2A–D). In addition, non-stimulated isogenic cell lines presented no differences regarding the expression of known CD44 interactors and downstream signaling partners (AKT, EGFR, ERK 1/2, P38 and STAT3, and respective phosphorylated forms) at the post-translational level (Figure 2E,F). In accordance with these results, tumor growth kinetics was also similar between MKN74_CD44v6 and MKN74_Mock cells (Figure 2G). The lack of an obvious correlation between overexpression of CD44v6 and the tested cancer hallmarks, suggests that CD44v6 is not a driver of GC progression.

### 2.2. CD44v6 Influences Response to Cisplatin Treatment in GC Cells

We then investigated whether CD44v6 influences response to conventional chemotherapeutic agents, namely cisplatin and 5-fluorouracil (5-FU), often included in the chemotherapy regimens provided to GC patients. We verified that MKN74_CD44v6 cells survive better to a clinically relevant cisplatin concentration (10 µM) (Figure 3A), compared to MKN74_Mock cells (*p* < 0.05), whereas no differential response was observed in response to 5-FU treatment (Figure 3B). Moreover, MKN74_CD44v6 cells also exhibited decreased apoptosis levels in response to cisplatin in comparison to both isogenic counterparts, *p* < 0.001 (Figure 3C). These results indicate that de novo expression of CD44v6 in GC cells allows tolerating cisplatin treatment. To confirm these results in a model that better mimics the natural context, we modeled the expression of CD44v6 in two non-edited GC cells that endogenously overexpress CD44v6 in all their cells (GP202 and MKN45; Appendix A) and treated them with cisplatin. Specific siRNAs were used to inhibit CD44v6 expression, with 50% and 90% inhibition levels being obtained for MKN45 and GP202, respectively (Figure 4A). CD44v6 depleted cells displayed higher cisplatin-induced apoptosis levels when compared to siRNA scramble controls, *p* < 0.001 (Figure 4B), indicating that CD44v6 expression inhibition renders cancer cells more sensitive to the effects of this drug. Although the siRNAs we used in this experiment can efficiently inhibit the expression of CD44v6, our results could have been reinforced if an additional set of CD44v6 siRNAs had also been used. Nevertheless, taken together, these results consistently indicate that, in these GC controlled models, CD44v6 modulates response to cisplatin treatment.

### 2.3. CD44v6-Induced Modulation to Cisplatin Response in GC Cells is Possibly Mediated by pSTAT3 or pP38

Since activation of STAT3 or P38 signaling pathways have been described as mediators of survival in response to cisplatin, we evaluated the levels of pSTAT3 and/or pP38 in the MKN74 isogenic model and in the GC cells that endogenously overexpress CD44v6, GP202 and MKN45 (Figure 4C–E). Regarding the isogenic model, none of the three cell lines present the phosphorylated form of STAT3, pSTAT3, at basal level (Figure 4D,E; and previously observed in Figure 2E). However, following 24 h of cisplatin treatment, pSTAT3 levels increased in MKN74_CD44v6 overexpressing cells, but also in MKN74_CD44s cells, but not in Mock cells (Figure 4D,E). Only after 48 h following cisplatin treatment, all three cell lines presented high pSTAT3 levels. Although STAT3 activation, through phosphorylation, becomes a general survival response after 48 h of cisplatin treatment, CD44v6 cells seem to more rapidly induce and maintain STAT3 activation. Regarding P38, the activated form of this protein is already present in the parental MKN74 cell line and no changes were observed in the remaining conditions (Appendix A). We then used GP202 and MKN45 GC cell lines (that endogenously express CD44v6) to test STAT3 and P38 activation following cisplatin treatment in CD44v6 depleted cells. As pSTAT3 is constitutive in both cell lines, we could not use it as a readout of CD44v6 overexpression, nor of cisplatin treatment (Appendix A). The CD44v6 expressing GP202 cells constitutively present pP38 (Figure 4C). Following cisplatin treatment, CD44v6 expressing cells show a small, although not statistically significant, increase in the activation of P38 (Figure 4C, upper left panel), without an evident difference in terms of cisplatin-induced apoptosis compared to vehicle control for this tested concentration (Figure 4B). However, upon CD44v6 depletion, pP38 expression is no longer observed in the nucleus of vehicle nor of cisplatin treated cells (Figure 4C, upper right panel), indicating that, in this cell line, nuclear pP38 expression is dependent on the presence of CD44v6. We can then hypothesize that the increased cisplatin sensitivity observed upon depletion of CD44v6 in GP202 cells, in Figure 4B, may be due to the loss of CD44v6 mediated pP38. MKN45 parental cells endogenously expressing CD44v6, also present P38 constitutively activated. Similarly to GP202, in the presence of CD44v6, there is only a small increase in P38 activation in response to cisplatin treatment (Figure 4C, lower left panel). However, contrary to GP202, in CD44v6 expressing MKN45 cells, there is a significant level of cisplatin-induced apoptosis, compared to vehicle (Figure 4B), which is probably related to the cisplatin concentrations used. Regarding P38 activation, while in GP202 cells this depends on the presence of CD44v6, in MKN45 cells P38 remains activated (and in the nucleus) of most cells upon CD44v6 depletion (Figure 4C, lower right panel). Interestingly, in MKN45 cells, CD44v6 also seems to be relevant for P38 activation, at least in the presence of cisplatin, since the percentage of cells with nuclear pP38 expression is significantly reduced when CD44v6 is depleted (Figure 4C, lower right panel). Nevertheless, in the presence of both CD44v6 and pP38, MKN45 cells cope better with cisplatin treatment than with only pP38. Overall, these results indicate that in the three GC cell lines, concomitant CD44v6 expression, STAT3 activation and P38 activation is associated with increased survival after cisplatin treatment.

### 2.4. CD44v6 Expressing Cells are Involved in GC Cell Overgrowth After Cisplatin Treatment

As CD44v6-positive (CD44v6+) cells seem to survive better after cisplatin treatment than CD44v6-negative cells (CD44v6−), it is likely that the proportion of CD44v6+ and CD44v6− cells in heterogeneous tumors (often found in GC, as shown in [29]) influences response to chemotherapy. We set out to test this hypothesis in vitro, by mimicking the effects of chemotherapy on a CD44v6 heterogeneous tumor. For that, we co-cultured CD44v6+ and CD44v6− cells (in a 50:50 ratio), treated them with cisplatin and then allowed them to recover for 15 days, without drug treatment. We then analyzed the proportion of the two cell populations at intermediate time-points and at the end of the experiment by flow cytometry (Figure 5A). During recovery from cisplatin treatment, and from day 9 onwards, the CD44v6+ cell population constitutes more than 70–80% of the entire cell culture, *p* < 0.0001 (Figure 5B). These results indicate that CD44v6 is involved in GC cell overgrowth after cisplatin treatment. If formally demonstrated in patients, this may suggest that cancer relapses after platinum-based chemotherapy may also be enriched in CD44v6+ cells.

## 3. Discussion

Aiming to unveil the role of CD44v6 in GC, we generated an isogenic model of GC cells stably expressing CD44s or CD44v6. Expression of CD44v6 or CD44s induced no differences in terms of doubling-time, invasion, cell-adhesion, cell motility, and tumorigenic potential in vivo. These experiments were performed in a tumorigenic GC cell line, which is a limitation of our study regarding possible conclusions on the capacity of CD44v6 de novo expression to influence gastric tumor initiation. Nevertheless, this lack of an obvious correlation between overexpression of CD44v6 and the tested cancer hallmarks, together with our earlier report showing that CD44v6 was already overexpressed in pre-malignant lesions [12] (most of which do not lead to cancer), suggests that CD44v6 is not a driver of GC progression.

Importantly, our data identify CD44v6 as a modulator of response to cisplatin treatment. Indeed, in the isogenic cell line model, overexpression of CD44v6 led to increased cell survival and decreased cisplatin-induced apoptosis. Likewise, downregulation of CD44v6-containing isoforms, in GC cell lines that endogenously express it, led to increased sensitivity to cisplatin-induced apoptosis. Similar results have been described in other cancer types [24,30], suggesting that expression of these variant isoforms could somehow mediate an escape mechanism to programmed cell death. In fact, expression of CD44v has been described to have a role in promoting chemoresistance through the upregulation of lyn kinase, via the Pi3K/AKT pathway in colon carcinoma cells [31] and CD44v6 in particular, was shown to block Fas mediated apoptosis [32]. The mechanism downstream of CD44v6, or cooperating with CD44v6, by which CD44v6 expressing cells have increased cisplatin survival is still unknown, however, our data show that in GC cells this may occur due to the concomitance of CD44v6 expression and activation of STAT3 or P38, depending on the cellular context. Indeed, high expression of activated STAT3 has been shown to contribute to cisplatin resistance in ovarian cancer [33]. Moreover, abrogation of STAT3 activity has been shown to circumvent cisplatin resistance in ovarian cancer cells [34], making it a promising target to reverse cisplatin resistance in ovarian and other cancer types [33,34,35]. It is therefore likely that the early activation of STAT3 we observed in response to cisplatin, in CD44-expressing cells, and mainly in CD44v6-expressing cells, is contributing to increase apoptosis resistance to this drug in the isogenic MKN74 cell line model. The mechanism by which CD44v6 is capable of modulating cisplatin response in GC cells seems to depend on the cellular context. In the two cell lines that endogenously express CD44v6 (MKN45 and GP202), it seems that activation of P38, which our data showed to be CD44v6 dependent in GP202 cells, is contributing to cisplatin resistance. Therefore, when we down-regulate the expression of CD44v6 in GP202 cells, P38 is no longer activated and cells become more sensitive to cisplatin. This is not surprising considering that inhibition of P38 has been shown to sensitize tumor cells to cisplatin-induced apoptosis [36]. In the MKN45 cells, where expression of pP38 is independent of CD44v6, it is not by losing pP38 that the CD44v6 depleted cells become more sensitive to cisplatin. This is probably due to another unexplored pathway. Nevertheless, what does become evident from our experiments is that both GP202 and MKN45 cells respond better to cisplatin treatment when they are expressing both CD44v6 and pP38. Indeed, the same can also be observed in the isogenic MKN74 cells that constitutively express pP38 (regardless of cisplatin treatment) but where the CD44v6 expressing cells cope better with cisplatin treatment. Regarding activation of STAT3, our data are less informative; however, as STAT3 becomes or remains activated upon cisplatin treatment in all tested cell lines, we may hypothesize that it also contributes, in a context of CD44v6 expression and P38 activation, to promote survival of cancer cells when these are challenged by cisplatin. Indeed, a regulatory mechanism between STAT3 and P38 has been described in head and neck cancer, whereby STAT3 phosphorylation requires P38 [37]; however, additional experiments would be required to assess if this is also the case in GC cells. Importantly our study has some limitations since it only provides indications that activation of STAT3 and P38 may be implicated in CD44v6 mediated response to cisplatin. To confirm these findings, additional studies would have to be performed using, for instance, STAT3 inhibitors to see if in their presence MKN74_CD44v6 cells regain sensitivity to cisplatin, equal to that observed in the MKN74_Mock cells. Interestingly, our experiments using co-cultures of CD44v6+ and CD44v6− cells show that CD44v6+ cells could have some selective advantage when a heterogeneous gastric tumor is treated with cisplatin-based chemotherapy. This selective advantage of CD44v6+ cells was only observed following 9 days of incubation with cisplatin, and appeared to be a sudden increase when compared with the previous time-point tested (6 days). Nevertheless, and considering those results represent the balance between the cells that are dying and the cells that are proliferating in each of the CD44v6+ and CD44v6− cell populations, we believe this was indeed a gradual increase in CD44v6+ cells that took place between day 6 and day 9. However, from the previous results described herein, we can assume that the molecular changes, that ultimately gave rise to this result, started to occur much earlier during/shortly after the cisplatin treatment. This overgrowth of CD44v6+ cells after chemotherapy, may indicate that this cell population can get enriched in GC relapses that arise after chemotherapy. This is plausible as CD44/CD44v are considered CSC markers in several tumor types, including GC [16,17]. Further studies in patients’ samples and supported by patients’ clinical data are required to address this subject. If our findings are confirmed, strategies to specifically eliminate CD44v6+ cells could lead to decreased recurrence and improved patient survival. One possible strategy is the targeting of CD44v6 downstream effectors, as direct targeting of CD44v6 ought to be avoided due to the lethal skin toxicity side effects described in a Phase I clinical trial using a highly potent antimicrotubule agent coupled to a monoclonal antibody against CD44v6 [38]. Our data also shed light into GC molecular heterogeneity and its relation with therapy, by suggesting that the CD44v6+ population is responsible for tumor overgrowth and may potentially help drive recurrence following conventional chemotherapy, in tumors composed of CD44v6+ and CD44v6− cell populations. Indeed, considering that more than half of GC tumors possess some degree of heterogeneity regarding CD44v6 expression (with CD44v6+ and CD44v6− cell populations co-existing in the same tumor) [29], this may have important consequences in terms of GC patients’ clinical outcome.

## 4. Materials and Methods

### 4.1. Cell Lines and Culture Conditions

Human GC cell lines MKN74 and MKN45 were purchased from the JCRB Cell Bank (Japanese Collection of Research Bioresources Cell Bank), and the non-commercial cell line GP202 cell line was established at Ipatimup [39]. All reagents were purchased from ThermoFisher Scientific (Waltham, MA, USA), unless otherwise stated. All cell lines were cultured in RPMI 1640 medium with 10% heat inactivated fetal bovine serum (FBS) (Biowest, Nuaillé, France). Cell lines were maintained at 37 °C and 5% CO_2_ in a high humidity atmosphere and sub-cultured every 3 to 4 days. Cells were grown in the absence of antibiotics except for cell selection in MKN74 cells, where G418 was used. Cells were never continuously cultured for more than 4 months. Cell identification was confirmed by STR analysis and cells were confirmed to be free of mycoplasma contamination.

### 4.2. Flow Cytometry

Flow cytometry was used to determine the expression of CD44 and/or CD44v6 in GC cell lines. All reagents were purchased from ThermoFisher Scientific (Waltham, MA, USA), unless stated otherwise. Cells were seeded in 6-well plates to a density of 5 × 10^5^ cells/well and allowed to grow to approximately 90% confluency. Cells were then washed with phosphate buffered saline (PBS), versinized for 5 min and washed in ice-cold PBS-3%BSA (bovine serum albumin). Cells were then blocked in PBS-3%BSA for 30 min and incubated with the appropriate primary antibody: mouse monoclonal antibody against CD44 (clone 156-3C11; 1:100 dilution; 60 min incubation; Cell Signaling Technology (Beverly, MA, USA)) or mouse monoclonal antibody against CD44v6 (MA54; 1:100; 60 min). Cells were subsequently washed with PBS-3%BSA and incubated with a secondary antibody: anti-mouse Alexa Fluor 488 or anti-mouse Alexa Fluor 564 (1:500; 60 min). Fluorescence was analyzed using a FACS Calibur or an Accuri C6 cytometer (both from BD Biosciences, Franklin Lakes, NJ, USA). The mean fluorescence intensity was measured for at least 20,000 gated events per sample and data were analyzed using the software Flow Jo, version 10.

### 4.3. RNA Extraction and cDNA Synthesis

All reagents were purchased from ThermoFisher Scientific (Waltham, MA, USA), unless otherwise stated. RNA was isolated using the mIRVana™ extraction Kit by following the supplier recommended protocol. Extraction efficiency and quality was assessed using a NanoDrop ND-1000. First Strand cDNA synthesis was performed using SuperScript^®^ II reverse transcriptase, random hexamer primers and 1 µg of template RNA, according to the manufacturers’ protocol.

### 4.4. Characterization of v6 Containing Transcripts in GP202 Cell Line

To characterize the full spectrum of v6 containing transcripts present in the GP202 cell line, we used a set of primers that amplifies from exon 5 through to exon 20. Forward primers included one overlapping exon 5 (primer A, Appendix A) and another overlapping exon v6 (primer B). A reverse orientation set of primers was designed to overlap exon v6 (primer D), exon boundary 17/18 (primer F), as well as two primers that amplify either the short tailed isoform variant, overlapping exon 19 (primer G) or the long tailed isoform variant, overlapping exon 20 (primer H). To identify the full length sequence of the v3–v10 transcript, we used a nested PCR approach to amplify the variable region of said transcript in the GP202 cell line. The first round of amplification included a forward primer overlapping exon 5 (primer A) and a reverse primer overlapping exon 20 (primer H). The second round of amplification was performed with a set of primers spanning the exon 5/exon v3 boundary (primer C, Appendix A) and the exon 16/exon 17 boundary (primer E). The resulting amplicons were resolved on a 3% agarose gel. All bands were excised and DNA extracted using a band extraction kit (GE Healthcare, Chicago, IL, USA). Sequencing was achieved with a primer walking strategy, with each primer designed to correctly map each inter-exon region between exons 5 and 20. The primers used for this task were primers, A, B, D, F, G and H (Appendix A). The sequence of this CD44v3–v10 (i.e., v6 containing) variant, endogenously expressed in the GP202 cell line, was then used to generate the isogenic cell model mentioned below.

### 4.5. Generation of An Isogenic Cell Line Model of Tumor CD44v6 Status

All reagents were from ThermoFisher Scientific (Waltham, MA, USA), unless otherwise stated. The sequences coding for CD44s (variant CD44-03-ENST00000263398) and CD44v6 (variant CD44-04-ENST00000415148; Ensembl release 68, July 2012) in a pCMV6-XL5 and pCMV6-AC backbone respectively, were purchased from OriGene. CD44s was excised with EcoRI and SmaI and cloned directly into a pIRES-EGFP2 plasmid. CD44v6 was excised in two steps, first with EcoRI and XhoI and sub-cloned into a pIRES-EGFP2 plasmid with a custom made MCS constructed by Bordeira Carriço et al. [40] and subsequently cleaved with SacI and SalI and subcloned into a pIRES-EGFP2 plasmid. Restriction enzymes were all purchased from New England Biolabs (Ipswich, MA, USA). The correct sequence of each transcript was confirmed by Sanger sequencing. The pIRES-EGFP2 empty vector was used as the mock control. The MKN74 cell line, was transfected using Lipofectamine 2000 reagent with either the empty pIRES-EGFP2 vector (MKN74_Mock), the pIRES-EGFP2_CD44v6 (MKN74_CD44v6) or the pIRES-EGFP2_CD44s (MKN74_CD44s). Selective pressure to isolate stably expressing cells, was applied with 1 mg/mL of G418, 48 h after transfection. In order to obtain pure populations of CD44v6 and CD44s cells, magnetic bead sorting with the Magnetic Separation kit CELLection Pan Mouse IgG kit from Dynabeads^®^ was performed, according to the manufacturers’ instructions. Briefly, following transfection, cells were sorted according to the expression of CD44v6 or CD44s. After the first round of separation, cells were allowed to grow to confluence, and two more rounds of magnetic bead sorting performed. The efficiency of cell selection was assessed by flow cytometry, as described above. Correct expression and localization, of the desired CD44 isoform, at the cell membrane was determined by immunofluorescence, as described below.

### 4.6. Immunofluorescence

All immunofluorescence reagents were from ThermoFisher Scientific (Waltham, MA, USA), and all antibodies were from Cell Signaling Technology (Beverly, MA, USA) unless stated otherwise.

Cells were grown on glass coverslips for the desired amount of time under normal growth conditions or with specific treatments before being washed twice in PBS and fixed with 4% paraformaldehyde (Merck, Darmstadt, Germany) for 20 min. Post fixation, cells were washed with PBS, incubated 10 min with NH_4_Cl, washed twice with PBS and incubated 5 min in PBS- 0.2%Triton X-100 for membrane permeabilization. Blocking was performed for 30 min with PBS-5% BSA. Coverslips were subsequently incubated with the desired primary antibodies. Detection of phosphorylated forms required methanol permeabilization for 10 min on ice prior to incubation with the primary antibody. Primary antibodies, diluted in PBS-5% BSA were: (i) mouse monoclonal antibodies–anti-CD44v6 (clone MA541; 1:100 dilution; 60 min incubation; ThermoFisher Scientific), anti-CD44 antibody (156-3C11; 1:100; 60 min), STAT3 antibody (124H6; 1:100); (ii) rabbit monoclonal antibodies–anti-P38 (D13E1; 1:100; ON incubation), anti-phospho-P38 (12F8; 1:100; ON), and phospho-STAT3 (D3A7; 1:100; ON). Cells were then washed twice with PBS and incubated in the dark with secondary antibodies: Alexa Fluor 488 Donkey Anti-Mouse secondary antibody (1:500; 60 min; Life Technologies, (Carlsbad, CA, USA) or Alexa Fluor 594 anti-rabbit (1:500, 60 min; Life Technologies), after which a final PBS wash was performed. In all cases, the cell nuclei were stained using Vectashield mounting media with DAPI (Vector Laboratories, Burlingame, CA, USA). Cells were analyzed by fluorescence microscopy (Imager.Z1, AxioCam fluorescence microscope or Eclipse TE-2000, both from Zeiss, Gottingen, Germany) using AxioVision software (Rockville, MD, USA).

### 4.7. Quantitative Real-Time Reverse Transcription PCR (qRT-PCR)

Gene expression levels were assessed by real-time qRT-PCR. RNA Isolation and cDNA synthesis was performed as described above. Real time qRT-PCR was performed in duplicate with 200 ng of template cDNA on an ABI Prism 7000 Sequence Detection System, using probes specific for CD44v6 (exon span V5–V6, Hs.PT.58.45400024) and total CD44 (exon span V2–V3, Hs.PT.58.4880087) (both from iDT, Corallvile, IA, USA). The relative expression level of CD44v6 and total CD44 was determined by the comparative 2^−ΔΔC_T_^ method using the housekeeping gene GADPH (4352934E; ThermoFisher Scientific, Waltham, MA, USA), to normalize expression results between samples.

### 4.8. Protein Extraction and Western Blot

Cells were cultured for the desired amount of time under normal growth conditions or with specific treatments before being washed twice with PBS and lysed with cold catenin lysis buffer [1% (*v/v*) Triton X-100, 1% (*v/v*) IGEPAL in PBS], supplemented with phosphatase (Sigma-Aldrich, Poole, UK) and protease (Roche, Basel, Switzerland) inhibitor cocktails. Total protein content of the samples was quantified by a modified Bradford Assay (Bio-Rad, Hercules, CA, USA). Thirty-five µg of total protein lysate were resolved by electrophoresis in a 7.5% or a 10% SDS polyacrylamide gel. The protein was transferred onto a Hybond nitrocellulose membrane (GE Healthcare, Chicago, IL, USA) and stained with Ponceau S solution (Sigma-Aldrich, Poole, UK) to confirm the efficacy of the transfer. Membrane blocking was performed by incubating membranes for 20 to 30 min with 5% non-fat milk diluted in 0.5% (*v/v*) PBS-Tween-20, or with 5% BSA diluted in 0.5% (*v/v*) TBS-Tween-20 or 5% non-fat milk diluted in 0.1% (*v/v*) TBS-Tween-20 (for the phosphorylated protein forms). Then, incubation of membranes with primary antibodies was performed overnight at 4 °C. Antibodies were all from Cell Signaling Technology (Beverly, MA, USA), unless stated otherwise. Primary antibodies used were: (i) mouse monoclonal antibody against CD44v6 (MA54; 1:500; ThermoFisher Scientific, Waltham, MA, USA), CD44 (156-3C11; 1:1000); (ii) rabbit monoclonal antibodies against AKT (1:1000), EGFR (D38B1; 1:1000), p44/42 MAPK (Erk1/2) (1:1000), P38 MAPK (D13E1; 1:1000), STAT3 (124H6; 1:1000), phospho-AKT (Ser473) (D9E; 1:1000), phospho-EGFR (D7A5; 1:1000), phospho-p44/42 MAPK (Thr202/Tyr204), (1:1000), phospho-p38 MAPK (Thr180/Tyr182) (12F8; 1:1000), phospho-STAT3 (Tyr705) (D3A7; 1:1000). α-tubulin (1:10,000; Sigma-Aldrich, Poole, UK) or α-actin (1:1000; Santa Cruz Biotechnology, Dallas, TX, USA) were used as loading controls. Membranes were further incubated with horseradish peroxidase-conjugated secondary antibodies (GE Healthcare and Santa Cruz) and labelled protein specific signal was detected by ECL (GE Healthcare, Chicago, IL, USA). Bands were quantified with the Quantity One software (Bio-Rad, Hercules, CA, USA). Protein for Western blotting was extracted from at least three independent biological replicates. Original Western blots are shown in Appendix A.

### 4.9. Cell Growth Assays

The Sulforhodamine B (SRB) assay or Presto Blue (PB; ThermoFisher Scientific, Waltham, MA, USA), a resazurin-based assay, were used to assess cell growth. All reagents were purchased from Sigma-Aldrich (Poole, UK) unless otherwise stated. Cells were seeded in 96-well plates (2.5 × 10^3^ per well) under normal conditions (5% CO2 humidified atmosphere at 37 °C) and allowed to adhere for 24 h. Cells were then allowed to grow for a further 48 h. At the required times: (i) for the SRB assay, cells were fixed in 10% trichloroacetic acid (TCA) for 1 h on ice, proteins stained with 4% SRB solution for 30 min, wells washed repeatedly with 1% acetic acid to remove the unbound dye, and the protein-bound stain was solubilized with 10 mM Tris solution. The SRB absorbance was measured at 560 nm and background corrected at 655 nm; (ii) for the PB assay, cells were incubated with PB solution (1×) for 45 min at 37 °C and fluorescence assessed (with an excitation wavelength of 560 nm and emission of 590 nm). Both absorbance and fluorescence were measured using a microplate reader (PowerWave HT Microplate Spectrophotometer; BioTek, Bad Friedrichshall, Germany). At least three independent experiments were performed, each measured in triplicate. The doubling time was calculated using the SRB assay as follows: Doubling Time = (0.301 × t)/(Log ODt/OD0), where ODt is the SRB absorbance at 48 h, OD0 is the SRB absorbance at an initial time and t is the time elapsed between the two (in this case, 48 h). Experiments to determine the IC50 for cisplatin were performed using the PB assay for all GC cell lines used (Appendix A): MKN74 IC50 ~ 3.6 μM; MKN45 IC50 ~ 6 μM and; GP202 IC50 ~ 17 μM. Cisplatin concentrations used in subsequent experiments were all performed within these ranges, depending on the assay. The selection of 5-FU concentrations used was performed according to Nakamura et al. [41].

### 4.10. Invasion and Slow Aggregation Assays

Cell invasion assays were carried out using matrigel invasion chambers (BD Biosciences, Franklin Lakes, NJ, USA), according to manufacturer´s specifications. Briefly, prior to cell seeding, membranes were hydrated on both sides with complete culture medium for 1 h at 37 °C. Cells were seeded (1 × 10^5^ per invasion chamber) and incubated for 24 h at 37 °C. Cells on the top portion of the filter were removed with a wet cotton swab. Invading cells, on the bottom portion of the membrane, were washed with PBS and fixed with ice cold methanol for 15 min. Each filter was detached from the plastic insert and mounted on a glass slide with mounting medium containing DAPI (Vector Laboratories, Burlingame, CA, USA). The total number of invading cells, from four independent biological replicates, was assessed by counting individual nuclei on a Zeiss Imager.Z1, AxioCam fluorescence microscope.

Slow aggregation assays, were performed by coating each well of a 96-well plate with 50 µL of a 6.7 mg/mL solution of bacto-agar in PBS. A total of 2 × 10^4^ cells were plated per well in triplicate for each experimental condition. Aggregation was monitored under an inverted microscope and photographed at 0, 24 and 48 h post seeding with a Nikon Digital Camera. At least three independent biological replicates were performed.

### 4.11. Wound Healing Assay

Wound healing assays were performed by seeding 2.1 × 10^4^ cells per condition in a culture insert µ-Dish (iBiDi, Martinsried Germany), according to manufacturers’ specifications. Briefly, after detachment of the cell culture insert, cell migration was monitored with a Nikon Digital Camera every 6 h. The last time point recorded was at 32 h, as the gap between the two migrating regions was already closed. At least three biological replicates were performed.

### 4.12. Tumor Growth Kinetics

All procedures involving animals were performed in accordance with the European Guidelines for the Care and Use of Laboratory Animals, Directive 2010/63/UE and the National Regulation published in 2013 (Law number: 113/2013, from August 7th) and approved by the national Ethics Committee from the Portuguese DGAV (ref. n° 013042/2017-05-08). The authors involved in these experiments have an accreditation for animal research given by the Portuguese National Authorities (Ministerial Directive 1005/92). NIH(S)II: nu/nu mice, a strain described by Azar et al. 1980 [42], were generated under Ipatimup/i3S supervision. MKN74_Mock and MKN74_CD44v6 cells (5 × 10^6^) were subcutaneously injected bilaterally into the back of four 6–8 week old male nude mice. Animals’ health was monitored daily. Tumor size was measured using external digital calipers, every 3 to 6 days, to assess tumor growth kinetics. Tumor volume was calculated using the following formula: length × width^2^ × 0.5. The experiment was terminated and mice humanely euthanized by cervical dislocation preceded by anesthesia with isoflurane when individual tumor volumes reached ~1000 mm^3^. Results are expressed as the average +/−SEM of three animals/tumors (each animal harbored one tumor from MKN74_Mock cells and one tumor from MKN74_CD44v6 cells).

### 4.13. Assessment of Cell Survival upon Drug Treatments

The effects of cisplatin and 5-FU (both from Sigma Aldrich, Poole, UK) on cell survival of the MKN74_Mock, MKN74_CD44v6 and MKN74_CD44s cells were evaluated by the PB assay, mentioned above. Briefly, cells were seeded in 96-well plates (2500 cells per well), allowed to adhere for 24 h and then incubated with cisplatin or 5-FU for 48 h. Cells were then processed for the PB assay, as described above. Cell survival for each drug treatment was calculated, as a percentage, in relation to the respective vehicle-treated control, for each cell line. At least three independent experiments were performed, each measured in triplicate.

### 4.14. Assessment of Cisplatin-Induced Apoptosis

MKN74_Mock, MKN74_CD44v6 and MKN74_CD44s cells were seeded in 6-well plates (1.0 × 10^5^ cells/ well), left to adhere for 24 h, and incubated with 10 µM of cisplatin for 24 and 48 h. Cisplatin-induced apoptosis was assayed by labelling cells with Annexin V-APC antibody (ImmunoTools, Friesoythe, Germany), according to the manufacturer’s instructions. Briefly, cells were trypsinized, pelleted, washed with PBS, resuspended in Annexin V binding buffer (10 mM HEPES, pH 7.4; 140 mM NaCl; 2.5 mM CaCl2) and incubated with Annexin V APC-conjugate for 15 min. Cells were then washed twice in PBS and resuspended in Annexin V binding buffer. Measurement of phosphatidylserine externalization was analyzed using an Accuri C6 flow cytometer and Accuri C6 software (BD Biosciences, Franklin Lakes, NJ, USA), plotting at least 20,000 events per sample. Results represent the average of at least three independent experiments.

### 4.15. CD44v6 Expression Inhibition by siRNA

GP202 and MKN45 cells were transfected with siRNAs for CD44v6 using Lipofectamine^®^ RNAiMax Transfection Reagent (ThermoFisher Scientific, Waltham, MA, USA), according to the manufacturers’ instructions. Briefly, cells were seeded in 6-well plates (2 × 10^5^ cells/well and 1.5 × 10^5^ cells/well, respectively GP202 and MKN45 cell lines). After 24 h, lipid based conjugates were prepared by mixing 20 nM of non-targeting siRNA (negative control DS NC1; iDT, Leuven, Belgium), or human CD44v6 siRNA (Sense strand: 5’-GCGUCAGGUUCCAUAGGAAUCCUTT-3’ and Antisense strand: 5’-AAAGGAUUCCUAUGGAACCUGACGCAG-3’, custom made from iDT, Leuven, Belgium), to diluted Lipofectamine^®^ RNAiMax Reagent in 1:1 ratio. The conjugates were incubated for 5 min at room temperature and added dropwise to the cells. Upon 48 h of incubation, protein extraction and Western blots were performed to evaluate the efficacy of CD44v6 silencing. In addition, in vitro cisplatin treatments (10 or 20 µM, for MKN45 and GP202 respectively) were carried out and apoptosis assessed (as described above).

### 4.16. Assessment of Percentage of CD44v6+ Cells upon Cisplatin Incubation

MKN74_Mock and MKN74_CD44v6 were mixed and seeded in 6-well plates with an initial density of 5 × 10^4^ cells/well (50:50 proportion), allowed to grow for 24 h and incubated with 10 μM of cisplatin or vehicle (0.9% (*v/v*) NaCl) for 6 h (preliminary experiments indicated 6 h cisplatin incubation to be suitable for subsequent long-term experiments, as it induced a reasonable degree of apoptosis and cells were still able to recover/proliferate a few days after cisplatin incubation). Cells were then washed with PBS, growth medium was replaced with cisplatin-free media and co-cultures were maintained during 15 days. CD44v6 cell enrichment was assessed at several time points by flow cytometry using a CD44v6 conjugated antibody. Briefly, at each time point, cells were washed with PBS, versinized (ThermoFisher Scientific, Waltham, MA, USA) for 15 min, blocked in PBS-2% FBS for 30 min and incubated with CD44v6-APC conjugated antibody (2F10; 10 μL/106 cells; 30 min; R&D Systems, McKinley Place, MN, USA). Cells were subsequently washed with PBS-2% FBS and analysed by flow cytometry. Fluorescence was analyzed using a FACS Canto II (BD Biosciences, Franklin Lakes, NJ, USA). The mean fluorescence intensity was measured for at least 20,000 gated events per sample and data were analyzed using the software FlowJo, version 10.

### 4.17. Statistical Analysis

For statistical significance assessment, a Two-way ANOVA with Tukey’s Post Hoc Test was performed. A *p*-Value < 0.05 was considered significant and all analyses were performed using GraphPad Prism version 8.2.1 for Windows.

## 5. Conclusions

In conclusion, we show that expression of CD44 isoforms containing exon v6 in the presence of activated STAT3 and P38, increase cell survival in response to cisplatin treatment in GC cells and that these cells override CD44v6-negative cells after cisplatin treatment. This suggests that tumor expression of CD44v6-containing variants may condition the outcome of GC patients treated with chemotherapy.

## Figures and Tables

**Figure 1 cancers-12-00858-f001:**
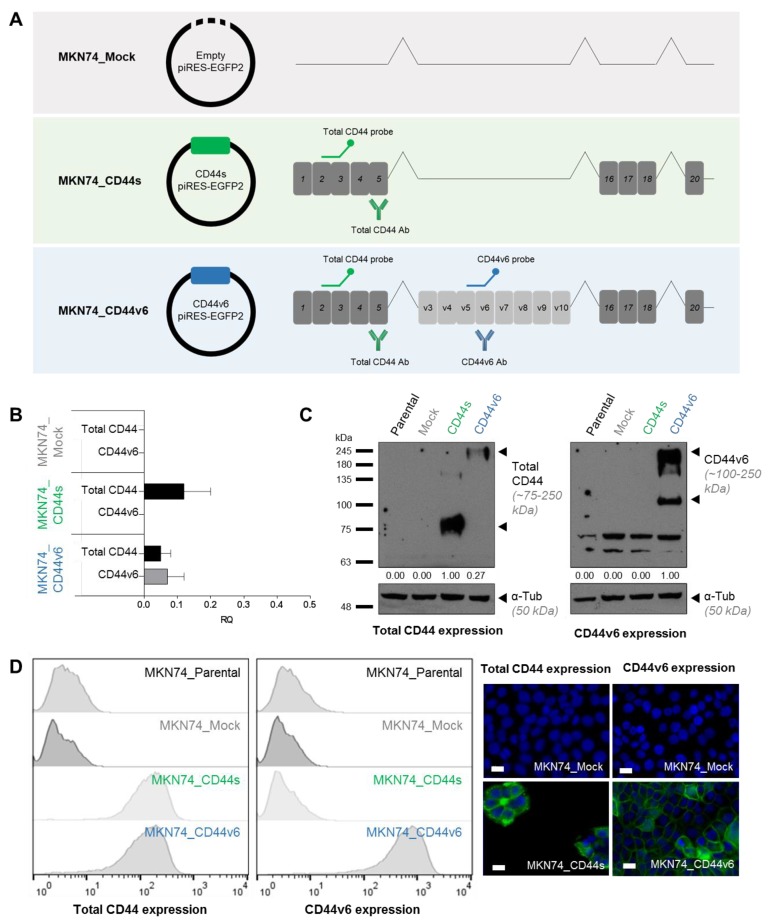
The generated MKN74 cells express the intended CD44 transcripts. (**A**) Transcripts transfected into the MKN74 (CD44v6 negative) gastric cancer (GC) cell line. The CD44 canonical form is the long cytoplasmic tail, exon 20 containing isoform (CD44-03 – ENST00000263398). The exon v6 containing transcript expresses the variable exons between v3-v10 (variant CD44-04 – ENST00000415148). Ensembl release 68, July 2012. It is highlighted where the probes and antibodies, used below, bind to the cDNA or protein, respectively; (**B**) Real time qRT-PCR representing the fold change expression of total CD44 and CD44v6 in transfected cells. Results are shown as average + SD and are representative of three independent experiments. The parental and MKN74_Mock transfected cells were used as negative controls; (**C**) Western Blot depicting total CD44 and CD44v6 protein levels in the generated MKN74 cells; (**D**) Expression was assessed at the post-translational level by immunofluorescence and flow cytometry against total CD44 and CD44v6. Nuclei are stained with DAPI (represented in blue) and white scale bars represent a distance of 20 µm.

**Figure 2 cancers-12-00858-f002:**
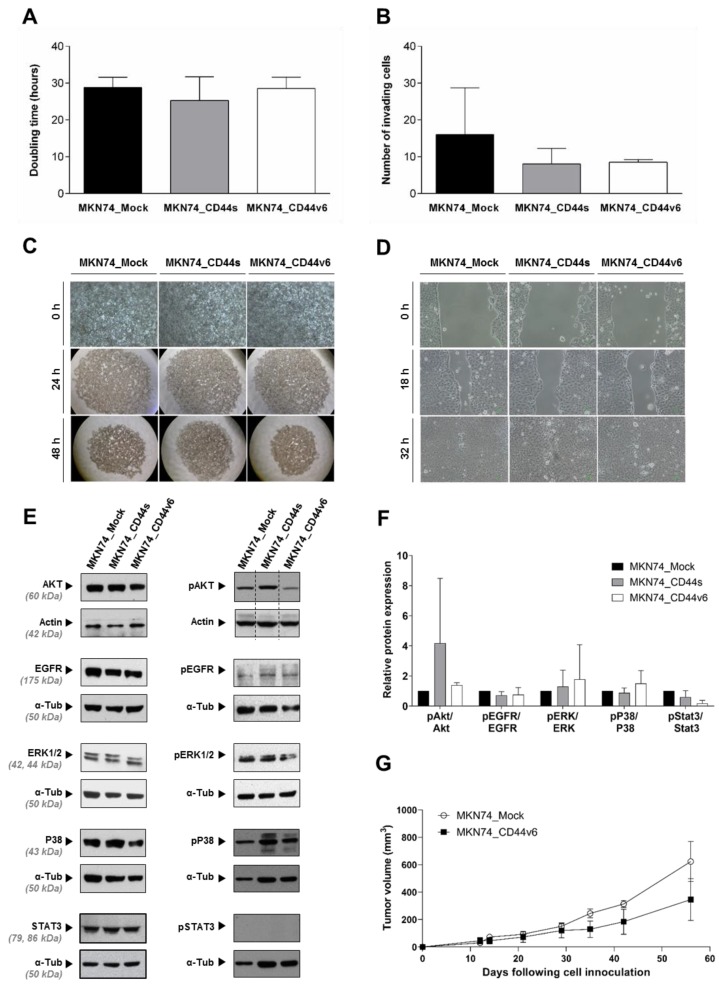
Functional characterization of the generated MKN74_Mock, MKN74_CD44s and MKN74_CD44v6 cells regarding: (**A**) Doubling time; (**B**) Invasion capacity; (**C**) Slow aggregation assay; (**D**) Migration capacity; (**E**) Protein expression of common CD44 interactors: AKT, EGFR, ERK 1/2, P38 and STAT3 and respective phosphorylated forms; (**F**) Relative protein quantification of pAKT, pEGFR, pERK 1/2, pP38 and pSTAT3 with respect to the respective total protein amount. All data are represented as average + SD and/or are representative of three independent experiments; (**G**) Tumor growth kinetics of MKN74_Mock and MKN74_CD44v6 xenografts in nude mice, presented as average +/− SEM. No statistical differences were observed.

**Figure 3 cancers-12-00858-f003:**
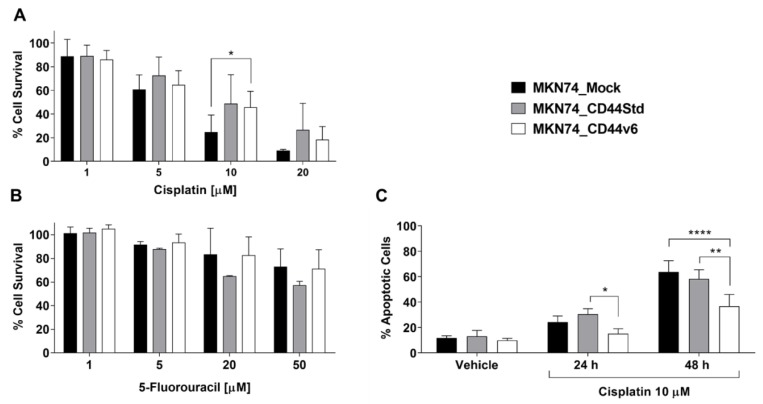
Assessing the response of the isogenic MKN74 cells to conventionally used chemotherapeutic agents: (**A**) Percentage cell survival upon incubation with cisplatin or (**B**) 5-FU for 48 h (compared to vehicle control) in MKN74 cells; (**C**) Percentage of apoptotic cells in MKN74 cells incubated with 10 µM cisplatin or vehicle (0.9% NaCl) for 48 h. Results are expressed as the average + SD of at least three independent experiments. Statistically significant results were determined by Two-way ANOVA with Tukey’s multiple comparisons test (* *p* < 0.05; ** *p* < 0.001; **** *p* < 0.0001).

**Figure 4 cancers-12-00858-f004:**
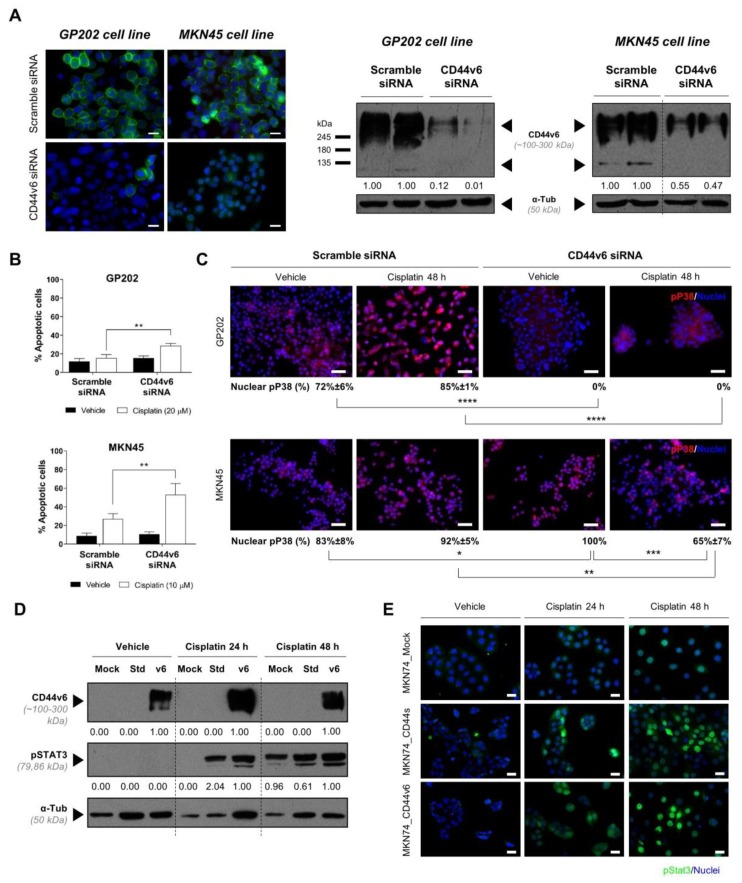
CD44v6-induced modulation to cisplatin response in GC cells is possibly mediated by pSTAT3 or pP38. (**A**) Immunofluorescence and Western blotting of CD44v6 in GP202 (left) and MKN45 (right) upon incubation with CD44v6 specific siRNAs, compared with incubation with scramble siRNA. Expression inhibition levels were, on average, ~90% and ~50% for GP202 and MKN45, respectively, in two independent experiments for each cell line. CD44v6 is represented in green; (**B**) Percentage of apoptotic cells in GP202 and MKN45 cell lines in response to 48 h treatment with cisplatin (concentrations selected according to Appendix A) or vehicle, following a 24 h incubation with scramble or CD44v6 siRNAs. Results are expressed as the average + SD of at least three independent experiments. Statistically significant results were determined by Two-way ANOVA with Tukey’s multiple comparisons test (* *p* < 0.05; ** *p* < 0.001; **** *p* < 0.0001); (**C**) Immunofluorescence of pP38 (seen in red) in GC cell lines treated with vehicle control or with cisplatin for 48 h (following a 24 h pre-incubation with scramble or CD44v6 siRNAs). Percentage of cells with nuclear pP38 expression is shown underneath the corresponding experimental conditions; (**D**) Western blotting of pSTAT3 in MKN74 isogenic cell lines in vehicle control and following 24 and 48 h with 10 µM cisplatin treatment. CD44v6 and pSTAT3 were run in the same gel against the same tubulin; (**E**) Immunofluorescence of pSTAT3 (seen in green) in vehicle and cisplatin treated cell lines. In all immunofluorescence images, nuclei are stained with DAPI (seen in blue) and white scale bars represent a distance of 50 µm.

**Figure 5 cancers-12-00858-f005:**
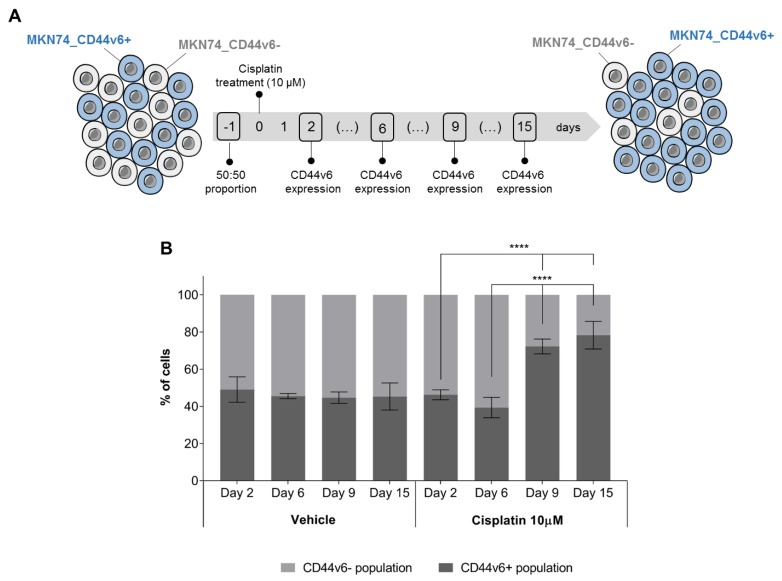
CD44v6+ expressing cells are involved in GC cell overgrowth after cisplatin treatment. (**A**) Scheme depicting the longitudinal assessment of CD44v6+ and CD44v6− population dynamics following cisplatin incubation, in a co-culture setting; (**B**) Percentage of CD44v6+ cells in co-cultures of CD44v6+ and CD44v6− MKN74 cells (MKN74_CD44v6 and MKN74_Mock cells, respectively) following a 6 h incubation with cisplatin (or vehicle control) and up to 15 days of recovery. At the end of the experiment > 80% of the cell culture is composed of CD44v6+ cells. The cisplatin concentration used is clinically relevant. Results expressed as the average +/− SD of at least three independent experiments. Statistically significant results were determined by Two-way ANOVA with Tukey’s multiple comparisons test (**** *p* < 0.0001).

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
