# Peer review of "Expression of CD44v6-Containing Isoforms Influences Cisplatin Response in Gastric Cancer Cells"

_cancers, 2020, doi:10.3390/cancers12040858_

Round 1

Reviewer 1 Report

There are currently hundreds of papers demonstrating CD44 expression and chemo-resistant cancers. Based on different contexts, their relationship could be positive or reverse. And this manuscript just brought us another piece of evidence for the “reverse” side. The authors mainly focused on one of CD44’s isoforms v6, whose inhibition leads to increase chemo-resistance to Cisplatin in GC.  To prove their hypothesis, the data they have presented was not convincing enough, and there are still some concerns await to be addressed:

  1. IC50 of each drug for each cell line to achieve the optimal concentrations for the experiment;
  2. From Fig. 2A, it seems that CDv44s also has a similar chemo-resistant effect at the dose of 20 μ Does it mean that at a higher dosage, both of the isoforms are resistant to the drugs?
  3. What is the concentration of the drug in Fig. 3C? 10 μM? Please label it clearly in the figure;
  4. It is obscure in Fig. 4A. Perhaps the authors would like to explain why they had two inhibition levels (I assume that they meant the cell density) since the manuscript does not mention the relationship between CD44 expression and the cellular density.
  5. Followed by the 4th concern, how did the authors achieve different inhibitive levels in the same set of experiments? By applying the same growing time with different amounts of siRNAs? Or different time with the same amount siRNAs? Or any other method?
  6. There is usually an off-target effect of siRNAs, it would be more convincing to use at least two in the same experiment;
  7. 4B is confusing. Are the “siRNA” groups were the same treatment? If those two cell lines are endogenously CD44v6 overexpressed, why did they use “siRNA” as a control?
  8. A qualified result (e.g. a bar chart) of Fig. 4C would be better;
  9. Were there any possible pathways involved in during this CD44v6-based chemo-resistant process? In the last part, the authors have found the activation of pSTAT3 and p38, but there was no further clue. Is it possible that there is a regulatory mechanism between these two?
  10. In Fig. 5B, the was 6 hrs incubation for the mixed cells. How was the time point identified? Was it according to the result of Fig. 2? (Still, preliminary data such as IC50 and time-dependent screening would be more convincing to identify the drug concentration and suitable time periods);
  11. In Fig. 4, the molecular alterations started in 48 hrs, however in Fig. 5B, there was no change until Day6, which was then followed by a dramatic increase in the CD44v6 population. How did that happen instantly instead of gradually? It could be discussed a bit.

Author Response

RESPONSE TO REVIEWER 1 COMMENTS

Thank you for your comments. We have addressed each one of them point-by-point by adding the requested information in the relevant sections of this manuscript. In addition, to address the raised issues, we have included additional analysis to some of our experiments and we prepared an additional Figure to include in the Supplementary File (Figure S3). At the end of the document, we also present a list of small changes that were introduced to the manuscript and we have also identified some inaccuracies in the manuscript, for which we would like to apologize, and for which we provide a list of all the corrections performed. All alterations to the original manuscript are shown in “track changes” and highlighted in yellow. We hope you find that we addressed your comments in a suitable manner and that this new version of the manuscript meets the expectations of the Reviewers and Editors and fulfills the criteria to be accepted for publication in Cancers.

Point 1: IC50 of each drug for each cell line to achieve the optimal concentrations for the experiment;

Response 1: Experiments to determine the IC50 for cisplatin (at 48 hours incubation) were performed for all cell lines, using a resazurin-based assay in 96 well plates. Results obtained were: for MKN74 IC50 ~ 3.6 μM, for MKN45 IC50 ~ 6 μM and for GP202 IC50 ~ 17 μM. We have included these results in the Supplementary File (Figure S3) and we now mention them in the Materials and Methods Section 4.9, Lines 422-424: “Experiments to determine the IC50 for cisplatin were performed using the PB assay for all GC cell lines used (Figure S3): MKN74 IC50~3.6 μM; MKN45 IC50 ~ 6 μM and; GP202 IC50 ~ 17 μM.”

For experiments in Figure 3A we performed the assays using a range of cisplatin concentrations around the previously determined IC50 for the parental MKN74 cell line. For the cisplatin-induced apoptosis experiment (Figure 3C) we used the 10 μM concentration since it seemed to be the more relevant bearing in mind the results in Figure 3A (i.e. 5 μM would be a very low dose to detect apoptosis in some of the isogenic cells, and 20 μM would probably be a very high dose for the Mock cells). For the experiments in Figure 4D and 4E, we used the same cisplatin concentration as in 3C.

For experiments in Figure 4B and 4C, we had to perform additional optimizations (within the IC50 range) to select a suitable concentration to detect cisplatin-induced apoptosis under the experimental conditions of that particular experiment. Therefore, for GP202 cells we performed preliminary studies using 10 and 20 μM and determined that 20 μM was the best concentration, since 10 induced very little apoptosis. Similarly, for MKN45 cells we treated cells with 5 and 10 μM, and determined that 10 μM was the best concentration to use. To include this information in the main text we added the following sentence in the Materials and Methods section 4.9, lines 424-425: “Cisplatin concentrations used in subsequent experiments were all performed within these ranges, depending on the assay.”

Regarding the 5-FU concentrations used in Figure 3B, we performed experiments using a range of concentrations around what is described in the literature as the IC50 for 5-FU for this cell line, 9.6 μM (according to Nakamura et al. 2014). We have now included this information (and respective reference) in the Materials and Methods section 4.9, line 425-426: “The selection of 5-FU concentrations used was performed according to Nakamura et al. [40]”.

Point 2: From Fig. 2A, it seems that CD44s also has a similar chemo-resistant effect at the dose of 20 μ. Does it mean that at a higher dosage, both of the isoforms are resistant to the drugs?

Response 2: We assume that Reviewer 1 is referring to Figure 3A. Indeed, when analyzing Figure 3A it does appear that at a higher concentration, both the CD44 expressing isoforms are more resistant to cisplatin, when compared to the Mock cells. However, we have rechecked our statistics and realized there was an error in the original graph as no statistically significant differences are detected between % cell survival at 20 µM for the three cell isoforms. There seems to be a tendency for CD44 expressing isoforms to be more resistant to cisplatin at 20 µM, but due to the variability between replicates, there was no statistical significance to confirm this. We apologize for the error in the original graph, which we have now corrected by including a new graph for Figure 3A.

Point 3: What is the concentration of the drug in Fig. 3C? 10 μM? Please label it clearly in the figure;

Response 3: The cisplatin concentration used in the experiments in Fig.3C was indeed 10 μM. We have now included this information in Figure 3C, as suggested by the Reviewer.

Point 4: It is obscure in Fig. 4A. Perhaps the authors would like to explain why they had two inhibition levels (I assume that they meant the cell density) since the manuscript does not mention the relationship between CD44 expression and the cellular density.

Response 4: Different CD44v6 inhibition levels (with RNAi) were obtained in GP202 and MKN45 cells. Expression inhibition in GP202 cells reached ~90%, whereas for MKN45 only a 50% inhibition was obtained, as assessed by Western blot, as stated in the figure legend. These different inhibition levels probably have to do with differences in transfection efficiency in both cell lines, and not a result of different cell density, since both cell lines were transfected at approximately the same density of 70%. However, when looking at the IF results (Figure 4A, left panel), it seems that almost all cells (from both cell lines) lost CD44v6 expression at the membrane, which reinforces our CD44v6 inhibition results.

Point 5: Followed by the 4th concern, how did the authors achieve different inhibitive levels in the same set of experiments? By applying the same growing time with different amounts of siRNAs? Or different time with the same amount siRNAs? Or any other method?

Response 5: The GP202 and MKN45 cells were seeded at different cell numbers (2x105 and 1.5x105, respectively), in order to achieve a similar cell density for transfection with siRNAs. Apart from that, all the other variables were the same between the two cell lines: the siRNA concentration and volume of transfection reagent (for which we performed preliminary experiments to determine the ideal siRNA concentration to use), as well as the growing time. As mentioned above, we believe these differences in inhibition levels have to do with the two cell lines having different transfection efficiencies.

Within the same cell line, the inhibition levels were equivalent between different experimental replicates, although this may not be very clear in line 116 of the main text. So the following modifications were included in lines 115-117 to correct this: “Specific siRNAs were used to inhibit CD44v6 expression with 50 and 90% inhibition levels being obtained for MKN45 and GP202, respectively (Figure 4 A).”

We also added to the legend of Figure 4 (lines 133-134) the following sentence “Expression inhibition levels were, on average, ~90% and ~50% for GP202 and MKN45, respectively, in two independent experiments for each cell line.”

Point 6: There is usually an off-target effect of siRNAs, it would be more convincing to use at least two in the same experiment;

Response 6: This is a relevant issue that points to the limitation of this data because we only used one set of siRNAs to target CD44v6, where ideally we should have used at least two sets of siRNA. We have addressed this in the Results section 2.2. in lines 120-122: “Although the siRNAs we used in this experiment can efficiently inhibit the expression of CD44v6, our results could have been reinforced if an additional set of CD44v6 siRNAs had also been used. Nevertheless, taken together….”

Point 7: 4B is confusing. Are the “siRNA” groups were the same treatment? If those two cell lines are endogenously CD44v6 overexpressed, why did they use “siRNA” as a control?

Response 7: From your comment, we realized that Figure 4B is confusing because it appears that “scramble” and “siRNA” (on the left-hand side) and “siRNA” and “CD44v6” (on the right-hand side) are different conditions. We have now changed the labelling of the X axis of those graphs to clarify this, and a new version of Figure 4B is presented.

Point 8: A qualified result (e.g. a bar chart) of Fig. 4C would be better;

Response 8: This is a relevant point that has also been raised by Reviewer 2. To address this issue, we have now included, in Figure 4C, the quantification of the % of cells with pP38 expression in the nucleus, as suggested by both Reviewers. An additional sentence was included in the legend of Figure 4 regarding this, in lines 141-142: “Percentage of  cells with nuclear pP38 expression is shown underneath the corresponding experimental conditions;” And this was also mentioned in the Results section, in lines 165 – 167: “…pP38 expression is no longer observed in the nucleus of vehicle nor of cisplatin treated cells (Figure 4C, upper right panel), indicating that, in this cell line, nuclear pP38 expression is dependent on the presence of CD44v6.”      

Point 9: Were there any possible pathways involved in during this CD44v6-based chemo-resistant process? In the last part, the authors have found the activation of pSTAT3 and p38, but there was no further clue. Is it possible that there is a regulatory mechanism between these two?

Response 9: There are several molecules known to modulate cisplatin response in cells, however we only explored whether the activation of STAT3 and P38 was being altered in the presence/absence of CD44v6, which we realize is a limitation of this study. 

Regarding STAT3 and P38, we found some indications that these molecules may be mediating the CD44v6-induced modulation of cisplatin response in our GC cells, however this appears to be dependent on the particular cellular context. It is indeed possible that there is a regulatory mechanism between those two molecules, in fact, it has been described that, in Head and Neck cancer, phosphorylation of STAT3 requires P38. In this work, we did not specifically investigate whether there was a regulatory mechanism between these two molecules but it is possible that is the case and further experiments are required to explore this, but that we believe are out of the scope of the present report.

We included a mention to this in the Discussion Section, lines 247 - 249: “Indeed, a regulatory mechanism between STAT3 and P38 has been described in head and neck cancer, whereby STAT3 phosphorylation requires P38 [37], however additional experiments would be required to assess if this is also the case in GC cells.” 

Point 10: In Fig. 5B, there was 6 hrs incubation for the mixed cells. How was the time point identified? Was it according to the result of Fig. 2? (Still, preliminary data such as IC50 and time-dependent screening would be more convincing to identify the drug concentration and suitable time periods);

Response 10: The choice of the 6 hour incubation with cisplatin resulted from preliminary experiments to optimize this. In general, the dose of cisplatin that cells receive depends on both the concentration of the drug and the duration of the treatment. In this experiment we wanted to use the 10 µM concentration (approximately the IC50 of the CD44 expressing isoforms and that was also the cisplatin concentration used in the apoptosis assay, Figures 3A and 3C). So we performed preliminary assays to adjust the incubation time for cisplatin in order to select an incubation period that would induce some level of cell death but without killing all cells or damaging them too much that they would not be able to recover from the cisplatin treatment and proliferate. We tested cisplatin incubation times between 1 hour and 48 hours, we then removed the cisplatin and incubated the cells in drug-free media for up to 15 days. We determined that, at this concentration, cells were not able to recover from cisplatin incubation times equal or above 24 hours, with almost all cells being dead after a few days. For the shorter incubation times, we considered that incubation times between 4 and 6 hours would be suitable for this experiment (as it induced a reasonable degree of apoptosis and still cells were able to recover/proliferate a few days after the incubation with cisplatin) and then selected the 6 hour incubation period to perform all subsequent experiments.

A mention to this issue was included in the Materials and Methods section 4.16, lines 499-501: “(preliminary experiments indicated 6 h cisplatin incubation to be suitable for subsequent long-term experiments, as it induced a reasonable degree of apoptosis and cells were still able to recover/proliferate a few days after cisplatin incubation)”

Point 11: In Fig. 4, the molecular alterations started in 48 hrs, however in Fig. 5B, there was no change until Day6, which was then followed by a dramatic increase in the CD44v6 population. How did that happen instantly instead of gradually? It could be discussed a bit.

Response 11: The results in Figure 5 represent the % of either CD44v6+ and CD44v6- cells in the population, which will be the result of the balance between the cells that are dying and the cells that are proliferating in each of those two cell populations. Indeed, from those results it looks that nothing was happening until after day 6 of the experiment and that there is then a sudden increase in the CD44v6+ population. We believe that the alterations are gradual and take place sometime between days 6 and 9. However, we did not analyse population representation between day 6 and day 9 of the experiment.

In addition, in this particular experiment we did not look at alterations at the molecular level, although our results suggest that these would likely start to occur in the first few hours after cisplatin treatment (as depicted in Fig. 4).

We have included this in the discussion, lines 257-264: This selective advantage of CD44v6+ cells was only observed following 9 days of incubation with cisplatin, and appeared to be a sudden increase when compared with the previous time-point tested (6 days). Nevertheless, and considering those results represent the balance between the cells that are dying and the cells that are proliferating in each of the CD44v6+ and CD44v6- cell populations, we believe this was indeed a gradual increase in CD44v6+ cells that took place between 6 and 9 days. However, from the previous results described herein, we may assume that the molecular changes, that ultimately gave rise to this result, started to occur much earlier during/shortly after the cisplatin treatment.

Additional modifications to the original manuscript:

  • CD44v6 expression inhibition in GP202 by RNAi was corrected to ~90%, instead of 80%, which was in the original version of this manuscript, in lines 116 and 133.

  • Included the information that cells in Figure 4D and 4E were treated with 10 μM cisplatin, in line 143.

  • Included details on the Presto Blue assay in the Materials and Methods (which was missing from the original text) in the (now re-named) section “4.9. Cell Growth assays”, from Line 407 to 417 and then in the section “4.13. Assessment of cell survival upon drug treatments” where we corrected that PB assay (and not SRB as was in the original version of the manuscript) was used to assess % cell survival: lines 465 and 467. Alterations are in track changes and highlighted in yellow.

  • Corrected the information regarding the concentration of CD44v6 siRNAs used, in Section 4.15 of Materials and Methods, line 488.

  • Included the information that cells in Figure 4B were treated with 10 or 20 μM cisplatin, for MKN45 and GP202 respectively, in Graph 4B and in line 494.

  • The Statistical Analysis was performed on GraphPad and not on SPSS as originally stated. This has now been corrected in line 513-514: “…all analyses were performed using GraphPad Prism version 8.2.1 for Windows.”

  • Formatted “exponential” numbers in lines: 294, 409, 431, 439, 444, 456, 460, 473, 486, and 498. Alterations are in “track changes” and highlighted in yellow.

  • Formatted words in “italics” in lines: 17, 47, 60, 111, 185, 206, and 494. Alterations are in “track changes” and highlighted in yellow.

  • Added 2 references (number 37 and 40) and re-numbered references from 37 onwards.

  • For all Western blot figures, densitometry readings/intensity ratio of each band were included (in Figures 1 and 4). In addition, the uncropped blots, showing all the bands with all molecular weight markers on the Western are presented in the Supplementary File 1.

  • Supplementary File 1 was also altered to include an additional Figure (Figure S3), requested by Reviewer 1, and subsequent Supplementary Figures were all re-numbered (both in the Supplementary File and in the main text).

  • Information regarding Supplementary Materials had to be updated to include a new Figure and the re-numbering of subsequent Figures. Lines 523 to 526.

  • In Figure 5 we found reasonable to change the color of MKN74_CD44v6 positive cells from reddish to blue to maintain the consistency of the color code attributed to each cell line, as illustrated in Figure 1A.

Reviewer 2 Report

In the manuscript by Pereira et al the authors examine if expression of CD44v6 alters cancer cell phenotypes and/or response to cisplatin treatment. CD44 variant isoforms are expressed in gastric cancer and associated with worse OS and potentially chemotherapy resistance increasing the relevance of the study. Overall the manuscript is well written. The majority of experiments are well-designed, controlled and analyzed with appropriate statistics and replicates. However, a few concerns exist as detailed below:

1) It is stated several times that because CD44v6 expression does not alter the cellular phenotypes tested that CD44v6 is not a driver of GC development. However, the cell line used is a GC cell line, already tumorigenic in vivo and already has high levels of phosphorylated p38 suggesting that it may rely on alternative pathways for its tumorigenic phenotype. To be able to make this claim CD44v6 should be expressed in a normal gastric epithelial cell line.

2) It would be good to compare the CD44 expression data in Figure 1B and C to one of the cell lines used in later figures that normally expresses CD44v6. Otherwise it is not clear if the CD44v6 levels are physiological

3) Figure S2 needs to be explained better. How it indicates CD44v6 expression in MKN74 cells is unclear (line 69).

4) Is CD44v6 expressed in all cells in the GP202 and MKN45 cell lines? It is hard to tell for sure from the IF data presented in Figure 4. If it is not 100%, it would affect the interpretation of other experiments.

5) The pP38 staining in figure 4 as presented is hard to interpret. The images do not appear to match the text. For example, the text states that there is no increase in pP38 in GP202 siRNA CD44v6 cells after platinum treatment (lines 157-158), but the red background in the image is very high. There also seems to be high levels of red background in scramble siRNA GP202 vehicle treated. Quantification of % positive cells and removal of background may help.

6) Overall the data presented in Figure 4 is not convincing. It is only correlative and does little to implicate pSTAT3 or pP38 in the effect of CD44v6 on response to platinum. The findings from these experiments are over-stated in the discussion (lines 222-226). Additional experiments should be performed to demonstrate the role of STAT3 and/or p38 in CD44v6’s role in platinum sensitivity.

7) The data presented in figure 5 is the most compelling piece of data in the manuscript. Demonstrating that phosphorylation of STAT3 or p38 is required for this result would greatly improve the impact of the study. Are all significant changes indicated in Figure 5B? The day 15 data also appear significant but is not indicated as such.

Author Response

RESPONSE TO REVIEWER 2 COMMENTS

Thank you for your comments. We have addressed each one of them point-by-point by adding the requested information in the relevant sections of this manuscript. In addition, to address the raised issues, we have included additional analysis to some of our experiments, we prepared an additional Figure to include in the Supplementary File (Figure S3), and we have also prepared Figures for your assessment (not for publication). At the end of the document, we also present a list of small changes that were introduced to the manuscript and we have also identified some inaccuracies in the manuscript, for which we would like to apologize, and for which we provide a list of all the corrections performed. All alterations to the original manuscript are shown in “track changes” and highlighted in yellow. We hope you find that we addressed your comments in a suitable manner and that this new version of the manuscript meets the expectations of the Reviewers and Editors and fulfills the criteria to be accepted for publication in Cancers.

Point 1: It is stated several times that because CD44v6 expression does not alter the cellular phenotypes tested that CD44v6 is not a driver of GC development. However, the cell line used is a GC cell line, already tumorigenic in vivo and already has high levels of phosphorylated p38 suggesting that it may rely on alternative pathways for its tumorigenic phenotype. To be able to make this claim CD44v6 should be expressed in a normal gastric epithelial cell line.

Response 1: Indeed, our study was performed in a tumorigenic GC cell line and not on a normal gastric epithelial cell line. This means that our study is limited regarding whether de novo expression of CD44v6 in normal gastric cells somehow contributes to the gastric tumor initiation process. Nevertheless, our results indicate that such a de novo expression in a gastric cancer cell does not lead to increased invasion, motility, doubling time nor tumorigenic potential, which are important for GC development. Moreover, in the literature there are very limited availability of “near-normal” cell lines that can reliably be used for this type of analysis.

We have included an additional sentence regarding this issue in the Discussion Section, lines 206-209:” These experiments were performed in a tumorigenic GC cell line, which is a limitation of our study regarding possible conclusions on the capacity of CD44v6 de novo expression to influence gastric tumor initiation. Nevertheless, this lack of….”   

Point 2: It would be good to compare the CD44 expression data in Figure 1B and C to one of the cell lines used in later figures that normally expresses CD44v6. Otherwise it is not clear if the CD44v6 levels are physiological

Response 2: This is a relevant point that we have now addressed by comparing protein expression between MKN74_CD44v6 and GP202 by Western blot (which we send for Reviewers assessment, not for publication: Figure 1 for assessment by Reviewer 2). CD44v6 expression levels are lower in MKN74_CD44v6 (0.68) than in GP202 cells (1.00), which can be expected because GP202 has several endogenously expressed CD44v6 containing isoforms (that will be recognized by the anti-CD44v6 antibody), whereas the MKN74_CD44v6 cell line only has the one CD44v6 isoform that we introduced in the vector. Nevertheless, considering the protein expression levels and the fact that CD44v6 is correctly expressed at the cell membrane (as shown in Figure 1 D), we can conclude that the CDv44v6 levels in the engineered MKN74_CD44v6 cell line are physiologically relevant. This was included in the main text in lines 74-75: “Moreover, the CD44v6 expression levels, obtained here, are physiologically relevant since they are similar to those endogenously expressed in the GP202 cell line (data not shown).”

Point 3: Figure S2 needs to be explained better. How it indicates CD44v6 expression in MKN74 cells is unclear (line 69).

Response 3: Your comment made us realize that the mention to Figure S2 in line 69 of the main text was indeed not clear. Figure S2 shows the identification of a CD44v6 containing isoform (CD44v3-v10) that we determined to be highly expressed in a gastric cancer cell line (GP202), leading us to select it to pursue our functional studies. We have now included the following text in lines 69 and 70 to clarify this: “… a CD44v6 containing isoform highly expressed in GC cells (GP202 cell line, Figure S2) [12], from now on mentioned as MKN74_CD44v6 cells (Figure 1A)]; …..”

Point 4: Is CD44v6 expressed in all cells in the GP202 and MKN45 cell lines? It is hard to tell for sure from the IF data presented in Figure 4. If it is not 100%, it would affect the interpretation of other experiments.

Response 4: CD44v6 is expressed in 100% of GP202 and MKN45 cells, although, as Reviewer 2 correctly mentions, this was not obvious from the IF presented in Figure 4A. We have now improved the definition of those images and we have also included additional information in Figure S1 that shows the analysis of expression of CD44v6 of both cell lines (MKN45 and GP202) by Flow cytometry, where it can be observed that 100% of cells express CD44v6. We also included a mention to this result in the main text - lines 114-115: “….that endogenously overexpress CD44v6 in all their cells (GP202 and MKN45; Figure S1)

Point 5: The pP38 staining in figure 4 as presented is hard to interpret. The images do not appear to match the text. For example, the text states that there is no increase in pP38 in GP202 siRNA CD44v6 cells after platinum treatment (lines 157-158), but the red background in the image is very high. There also seems to be high levels of red background in scramble siRNA GP202 vehicle treated. Quantification of % positive cells and removal of background may help.

Response 5: Indeed, the images shown in Figure 4C are not easy to interpret due to the small size of the images and the existence of some background. We have now removed the background from those images and have included a quantification of the % of cells with pP38 expression in the nucleus, as suggested by both Reviewers. We believe this now clarifies our interpretation of the obtained results, regarding there being no increase in pP38 in GP202 cells with depleted CD44v6 upon cisplatin treatment (lines 165-167 of the revised version). Indeed, what appears to be a high background in that image is probably an increased accumulation of pP38 in the cytoplasm without it going to the nucleus. This may still be difficult to observe even after background correction, so we are also providing better quality images with the individual fluorescence channels, for Reviewer 2 to assess (Figure 2 for assessment by Reviewer 2). These results also became clearer upon quantification of the % of cells with pP38 in the nucleus. Also to clarify this, we have included a mention to the nuclear expression of pP38 in the nucleus of these cells, in the main text, lines 165 to 167: “…pP38 expression is no longer observed in the nucleus of vehicle nor of cisplatin treated cells (Figure 4C, upper right panel), indicating that, in this cell line, nuclear pP38 expression is dependent on the presence of CD44v6.”       

Point 6: Overall the data presented in Figure 4 is not convincing. It is only correlative and does little to implicate pSTAT3 or pP38 in the effect of CD44v6 on response to platinum. The findings from these experiments are over-stated in the discussion (lines 222-226). Additional experiments should be performed to demonstrate the role of STAT3 and/or p38 in CD44v6’s role in platinum sensitivity.

Response 6: Results presented in Figure 4 only provide some clues that activation of STAT3 or of P38 may be mediating the CD44v6-induced modulation of cisplatin response in our GC cells, depending on the cellular context. These results make sense considering what is described in the literature, nevertheless we are aware that the conclusions we can take from these results are limited and that it would be required to perform additional studies to confirm it is via these molecules that CD44v6 is mediating cisplatin response. This could be performed with STAT3 inhibitors, for instance, to see if in their presence the MKN74_CD44v6 cells gain sensitivity to cisplatin. Such experiments, although important, are beyond the scope of this manuscript.

To include this important issue in the discussion, we included the following sentence in lines 249 - 254: “Importantly our study has some limitations since it only provides indications that activation of STAT3 and P38 may be implicated in CD44v6 mediated response to cisplatin. To confirm these findings, additional studies would have to be performed using, for instance, STAT3 inhibitors to see if in their presence MKN74_CD44v6 cells re-gain sensitivity to cisplatin, equal to that observed in the MKN74_Mock cells.”  

Point 7: The data presented in figure 5 is the most compelling piece of data in the manuscript. Demonstrating that phosphorylation of STAT3 or p38 is required for this result would greatly improve the impact of the study. Are all significant changes indicated in Figure 5B? The day 15 data also appear significant but is not indicated as such.

Response 7: Thank you for your comment. We did not assess activation of STAT3 or of P38 in the experiments presented in Figure 5, but from experiments shown in Figure 4D and E, we believe that both cell lines used (MKN74_mock and MKN74-CD44v6 cells) would have activation of STAT3 at the 48 h time-point. Assessing STAT3 activation throughout the experiment could have provided us with additional clues whether it is the early activation of STAT3 in MKN74_CD44v6 cells that is making them more resistant to cisplatin, especially if this experiment was coupled with a STAT3 inhibitor. Unfortunately these are experiments that would take several weeks to perform, including all replicates necessary, so unfortunately we do not have sufficient time to perform them at the moment, to comply with the journal request for re-submission.

We have now highlighted all significant changes in Figure 5B.

Additional modifications to the original manuscript:

  • CD44v6 expression inhibition in GP202 by RNAi was corrected to ~90%, instead of 80%, which was in the original version of this manuscript, in lines 116 and 133.

  • Included the information that cells in Figure 4D and 4E were treated with 10 μM cisplatin, in line 143.

  • Included details on the Presto Blue assay in the Materials and Methods (which was missing from the original text) in the (now re-named) section “4.9. Cell Growth assays”, from Line 407 to 417 and then in the section “4.13. Assessment of cell survival upon drug treatments” where we corrected that PB assay (and not SRB as was in the original version of the manuscript) was used to assess % cell survival: lines 465 and 467. Alterations are in track changes and highlighted in yellow.

  • Corrected the information regarding the concentration of CD44v6 siRNAs used, in Section 4.15 of Materials and Methods, line 488.

  • Included the information that cells in Figure 4B were treated with 10 or 20 μM cisplatin, for MKN45 and GP202 respectively, in Graph 4B and in line 494.

  • The Statistical Analysis was performed on GraphPad and not on SPSS as originally stated. This has now been corrected in line 513-514: “…all analyses were performed using GraphPad Prism version 8.2.1 for Windows.”

  • Formatted “exponential” numbers in lines: 294, 409, 431, 439, 444, 456, 460, 473, 486, and 498. Alterations are in “track changes” and highlighted in yellow.

  • Formatted words in “italics” in lines: 17, 47, 60, 111, 185, 206, and 494. Alterations are in “track changes” and highlighted in yellow.

  • Added 2 references (number 37 and 40) and re-numbered references from 37 onwards.

  • For all Western blot figures, densitometry readings/intensity ratio of each band were included (in Figures 1 and 4). In addition, the uncropped blots, showing all the bands with all molecular weight markers on the Western are presented in the Supplementary File 1.

  • Supplementary File 1 was also altered to include an additional Figure (Figure S3), requested by Reviewer 1, and subsequent Supplementary Figures were all re-numbered (both in the Supplementary File and in the main text).

  • Information regarding Supplementary Materials had to be updated to include a new Figure and the re-numbering of subsequent Figures. Lines 523 to 526.

  • In Figure 5 we found reasonable to change the color of MKN74_CD44v6 positive cells from reddish to blue to maintain the consistency of the color code attributed to each cell line, as illustrated in Figure 1A.

Round 2

Reviewer 1 Report

I think the authors have addressed my concerns, thus I recommend it for publication.

Author Response

Thank you for recommending our manuscript for publications. We are pleased to have addressed your comments in a suitable manner and that the new version of the manuscript met your expectations.

Reviewer 2 Report

In the revised manuscript many concerns from the original review were addressed but several still remain.

1) In the abstract, line 23-24 states that “suggesting CD44v6 is not a driver of GC 24 development”. Since CD44v6 was not tested in normal cells this is not an accurate statement and should be tempered to reflect that CD44v6 expression in a cancer cell line does not alter cancer cell phenotypes. The phrase “GC development” is used throughout the text and is misleading. GC progression may be more appropriate.  

2) Furthermore, in line 24-25 it is indicated that “Upon cisplatin treatment, CD44v6+ cells survive better and have lower apoptosis levels than CD44v6- cells, likely due to concomitant activation of STAT3 and P38”  and lines 27-28 “we show that CD44v6 expression, in the presence of activated STAT3 and 28 P38, increases cell survival in response to cisplatin treatment in GC cells”. It was suggested in the previous review that the author include further experiments to investigate the connection to STAT3 and p38, which they chose not to do. In the current form the data presented is not strong enough to support any connection between CD44v6 and STAT3 and/or p38 activation. The pSTAT3 data presented in figure 4D&E shows an earlier increase in pSTAT3 levels in cells expressing both CD44v6 and CD44s. Because CD44s expression does not also significantly decrease cell survival after platinum treatment in Figure 3, the connection between pSTAT3 and cell survival is not clear.  In regard to p38, p38 is expressed in MKN45 and is not altered by platinum or CD44v6 expression. However, CD44v6 expression alters levels of platinum-induced apoptosis. In GP202 cells depletion of CD44v6 results in decrease in nuclear p38. In MKN45 there is no change in p38 levels upon CD44v6 depletion. Since there is no consistent connection between these factors in the different cell lines, it is not appropriate to speculate that the change in p38 levels in CD44v6 depleted cells has a role in the apoptotic response without experimental evidence.

3) Figure 1 provided for the reviewers is supposed to demonstrate that CD44v6 is expressed at physiological levels similar to the levels in GP202. However, different blots are presented for the different cell lines. If the protein lysate is not run on the same blot, levels of expression between the cell lines cannot be compared. Also, once completed correctly it is unclear why this data is not included in the manuscript.

4) The authors added the percentage of p38 positive cells to Figure 4C, which is helpful in interpreting the data. However, this data should be presented as a mean +/- error to display the variability between biological replicates. The authors claim that there is in an increase nuclear p38 after platinum treatment in GP202 and MKN45 cells, but this needs to be supported by appropriate statistics.  

Author Response

Thank you for your comments. We have addressed each one of them point-by-point by adding the requested information in the relevant sections of this manuscript. All alterations to the original manuscript are shown in “track changes” and highlighted in yellow. We hope you find that we addressed your comments in a suitable manner and that this new version of the manuscript meets the expectations of the Reviewers and Editors and fulfills the criteria to be accepted for publication in Cancers.

1) In the abstract, line 23-24 states that “suggesting CD44v6 is not a driver of GC 24 development”. Since CD44v6 was not tested in normal cells this is not an accurate statement and should be tempered to reflect that CD44v6 expression in a cancer cell line does not alter cancer cell phenotypes. The phrase “GC development” is used throughout the text and is misleading. GC progression may be more appropriate. 

Point 1: We agree with the point raised by Reviewer 2 and have, therefore, altered the text accordingly by replacing the word “development” with “progression” in lines 24, 67, 84 and 218,  

2) Furthermore, in line 24-25 it is indicated that “Upon cisplatin treatment, CD44v6+ cells survive better and have lower apoptosis levels than CD44v6- cells, likely due to concomitant activation of STAT3 and P38” and lines 27-28 “we show that CD44v6 expression, in the presence of activated STAT3 and 28 P38, increases cell survival in response to cisplatin treatment in GC cells”. It was suggested in the previous review that the author include further experiments to investigate the connection to STAT3 and p38, which they chose not to do. In the current form the data presented is not strong enough to support any connection between CD44v6 and STAT3 and/or p38 activation. The pSTAT3 data presented in figure 4D&E shows an earlier increase in pSTAT3 levels in cells expressing both CD44v6 and CD44s. Because CD44s expression does not also significantly decrease cell survival after platinum treatment in Figure 3, the connection between pSTAT3 and cell survival is not clear. In regard to p38, p38 is expressed in MKN45 and is not altered by platinum or CD44v6 expression. However, CD44v6 expression alters levels of platinum-induced apoptosis. In GP202 cells depletion of CD44v6 results in decrease in nuclear p38. In MKN45 there is no change in p38 levels upon CD44v6 depletion. Since there is no consistent connection between these factors in the different cell lines, it is not appropriate to speculate that the change in p38 levels in CD44v6 depleted cells has a role in the apoptotic response without experimental evidence.

Point 2: We agree that line 24-25 and 27 and 28 of the abstract may appear to confer too much certainty regarding the importance of STAT3 and P38 in mediating response to cisplatin in CD44v6 expressing GC cells. This was not our intention, and we altered the sentences accordingly: i) in line 25, “….CD44v6+ cells survive better and have lower apoptosis levels than CD44v6- cells, possibly due to concomitant activation of STAT3 and P38”; ii) in lines 27-28, to: “In conclusion, we show that CD44v6 expression increases cell survival in response to cisplatin treatment in GC cells and that these cells override CD44v6-negative cells after cisplatin-treatment.” and, iii) to highlight this, we also replaced “likely” with “possibly”, in lines 131, 148, throughout the text, and altered line 229-230: “….that in GC cells this is may occur due to the concomitance of CD44v6 expression and by the activation…”

We understand that it would have been important to include additional experiments to confirm that STAT3 and P38 are indeed involved in CD44v6 mediated cisplatin response. However, it is not possible for us to perform additional experiments on this subject in the required 10 days, to comply with the journal request for re-submission (Due to a COVID-19 case in our institute, it has been closed since the 13th March till at least the 31th March).

We are aware that we only have an indication that STAT3 and/or P38 activation may be involved in cisplatin response mediated by CD44v6. We are also aware that this association may be different for all the cell lines tested. However, we believe that the pathways by which CD44v6 mediates its cisplatin response depend on the cellular context, as mentioned in lines 231 and 238. Herein, we just wish to highlight that in all three cell lines, the context that promotes survival in the presence of cisplatin always display concomitantly expression of CD44v6, activation of P38 and/or activation of STAT3, independently if this is promoted by CD44v6 or pre-existing in the cells. Nevertheless, we agree with Reviewer 2 and acknowledge that our study has limitations regarding the conclusions we can obtain regarding STAT3 and P38. Therefore, we addressed this topic in the Discussion section of the manuscript during Round 1 of the revision (in lines 256 – 260).

3) Figure 1 provided for the reviewers is supposed to demonstrate that CD44v6 is expressed at physiological levels similar to the levels in GP202. However, different blots are presented for the different cell lines. If the protein lysate is not run on the same blot, levels of expression between the cell lines cannot be compared. Also, once completed correctly it is unclear why this data is not included in the manuscript.

Point 3: The reviewer is absolutely correct. Indeed these proteins were run in different gels, and although both gels were run together, it is difficult to compare expression levels between both samples. We intended to perform new Western blots to obtain an accurate comparison of CD44v6 expression between MKN74_CD44v6 cells and GP202 or MKN45 cells. However our Research Institute has now been closed since the 13th of March (and its researchers in isolation at home) due to the Covid-19 pandemic and national emergency going on at the moment, and we do not know when this situation will be resolved. Although we do not have accurate protein expression levels from a Western blot (and will not be able to produce one in the allocated 10 days), we believe that the Immunofluorescence data, presented in Figure 1D shows that MKN74_CD44v6 cells have a strong expression of CD44v6 and that this is well localized in the membrane, similar to what is observed in cells GP202 and MKN45 (that endogenously express CD44v6) in Figure 4A, indicating that the CD44v6 levels in the engineered MKN74_CD44v6 cells are physiologically relevant.

4) The authors added the percentage of p38 positive cells to Figure 4C, which is helpful in interpreting the data. However, this data should be presented as a mean +/- error to display the variability between biological replicates. The authors claim that there is in an increase nuclear p38 after platinum treatment in GP202 and MKN45 cells, but this needs to be supported by appropriate statistics. 

Point 4: We agree with Reviewer 2 and have now altered Figure 4C to present the data as mean +/- SD of the biological replicates. By analyzing these results we noted that, although there is a small increase in the percentage of GP202 and MKN45 cells with nuclear pP38 in the presence of cisplatin (in the scramble siRNA condition), this was not statistically significant and have therefore included this information in the text in line 165, for GP202: “….show a small, although not statistically significant, increase in the activation of P38…”; and in line 173, for MKN45: “…in the presence of CD44v6, there is only a small increase in P38 activation…”

In addition, we also altered the subsequent sentences accordingly, lines 177-179: “Regarding P38 activation, while in GP202 cells this depends on the presence of CD44v6 expression, in MKN45 cells P38 remains activated (and in the nucleus) even upon CD44v6 depletion…” and in line 243 of the Discussion Section: “In the MKN45 cells, where the expression of pP38 is independent of CD44v6, it is not by….”

Also, we noted that there was a statistically significant reduction in the percentage of nuclear pP38 in cisplatin treated and CD44v6 depleted MKN45 cells. We included this information in lines 180-183: “Interestingly, in MKN45 cells, CD44v6 also seems to be relevant for P38 activation, at least in the presence of cisplatin, since the percentage of cells with nuclear pP38 expression is significantly reduced when CD44v6 is depleted (Figure 4C, lower right panel).“

Round 3

Reviewer 2 Report

The authors have adequately addressed the majority of this reviewer's concerns. It is acknowledged that the concern about the expression level of CD44v6 in MKN74_CD44v6 versus GP202 cells cannot be addressed at this time. This concern is not significant enough to prevent publication of this manuscript.